# The Role of Biomarkers in Surgery for Ulcerative Colitis: A Review

**DOI:** 10.3390/jcm10153362

**Published:** 2021-07-29

**Authors:** Jared Matson, Sonia Ramamoorthy, Nicole E. Lopez

**Affiliations:** Department of Surgery, UC San Diego, San Diego, CA 92037, USA; jsmatson@health.ucsd.edu (J.M.); sramamoorthy@health.ucsd.edu (S.R.)

**Keywords:** biomarkers, ulcerative colitis, surgery, inflammatory bowel disease, colorectal surgery

## Abstract

Ulcerative colitis (UC) is an inflammatory condition that generally affects the rectum and extends proximally into the colon in a continuous, distal-to-proximal pattern. Surgical resection (total proctocolectomy) is the only cure for UC and is often necessary in managing complicated or refractory disease. However, recent advances in biologically targeted therapies have resulted in improved disease control, and surgery is required in only a fraction of cases. This ever-increasing array of options for medical management has added complexity to surgical decision-making. In some circumstances, the added time required to ensure failure of medical therapy can delay colectomy in patients who will ultimately need it. Indeed, many patients with severe disease undergo trials of multiple medical therapies prior to considering surgery. In severe cases of UC, continued medical management has been associated with a delay to surgical intervention and higher rates of morbidity and mortality. Biomarkers represent a burgeoning field of research, particularly in inflammatory bowel disease and cancer. This review seeks to highlight the different possible settings for surgery in UC and the role various biomarkers might play in each.

## 1. Introduction

Ulcerative colitis (UC) is an idiopathic, chronic, relapsing-remitting inflammatory condition that typically affects the rectum and colon in contiguous fashion [1]. The latter helps to distinguish it from Crohn’s Disease (CD), another type of inflammatory bowel disease (IBD), which most often afflicts the terminal ileum, but may appear anywhere from mouth to anus. The incidence of UC varies between 6 and 24 per 100,000 persons per year, with higher incidence found in western countries and prevalence as high as 0.6% in Canada [2]. Moreover, the incidence appears to be rising worldwide and be associated with increasing industrialization [2]. This represents a significant burden on the healthcare system, as patients with UC tend to utilize healthcare resources at notably higher rates than other patients [3].

Major advances in medical therapy, particularly the advent of biologics in the last 20 years, have reduced the rates of ED visits, hospitalization, and short-term surgery [4,5,6]. Patients may initially be treated with aminosalicylates, corticosteroids, and immune modulators. Nonresponders and patients presenting with severe disease often receive biologic agents (and may even receive second- or third-line biologic agents) before surgery is considered. Still, 10–30% of patients admitted with acute severe colitis (ASC, generally defined by Truelove and Witts criteria as ≥6 bloody bowel movements per day as well as any of the following: tachycardia ≥90 bpm, fever >37.8 °C, anemia with hemoglobin <10.5 g/dL, and/or an ESR >30 mm/h) will require short-term colectomy (variably defined in studies as surgery during the index admission to up to 90 days from initial hospitalization) [7,8,9,10,11].

A proposed benefit of extended trials of medical therapy is to reduce the need for urgent or emergent colectomy, which has worse outcomes than elective operations [12]. However, these trials require time to determine effectiveness of therapy and patients who experience a delay in surgery also have worse outcomes (including increased mortality rates and intra- and post-operative complications such as hemorrhage, wound infection, enterocutaneous fistula, myocardial infarction, and need for reoperation) [12,13,14]. Accordingly, the American Society of Colon and Rectal Surgery (ASCRS) guidelines recommend 48–96 h to determine response to steroids and another five to seven days to assess for efficacy of biologics or cyclosporine [15], while the Italian Society of Colorectal Surgery (SICCR) recommends beginning “rescue” therapy after three days of steroid nonresponsiveness, and allowing seven days for improvement on a second-line agent [16]. Unfortunately, several studies show that delays as short as three days can increase morbidity and mortality if medical therapy fails and patients require an operation [13,14]. Therefore, early identification of patients who will need surgery is essential.

While patients with perforation, life-threatening bleeding, severe systemic illness, and multisystem organ dysfunction clearly require emergent surgical intervention, in other cases, it is often less obvious who will need an operation and when. Various biomarkers have been used to assist in distinguishing IBD from other gastrointestinal disorders and predicting relapse for IBD patients in remission, among other applications. Some biomarkers have been linked to surgical decision making with the potential to predict failure of medical treatments. However, the precise role of biomarkers in determining which patients will need surgery and when they should receive an operation is still evolving. Here the authors perform the first review in the published literature specifically examining the role of biomarkers in determining the need for surgery in UC.

## 2. Principles of Surgery for UC

Though there have been significant strides made in medical treatments, surgery remains the only cure for UC, solidifying the surgeon’s role as essential in the multidisciplinary care of patients with UC [17,18,19]. Indications for surgical management include stabilizing patients with life-threatening pathology, managing symptoms in refractory disease, preventing severe complications, and providing an acceptable quality of life. It is helpful to conceptualize surgery for UC in terms of three phases: emergent, urgent, and elective operations. Currently, there are no biomarkers to assist in decision-making in emergency circumstances. However, the decision to proceed to surgery in emergencies should be a purely clinical decision and waiting for a biomarker to confirm that decision would be both senseless and would, by definition, contradict the nature of categorizing the intervention as emergent. In contrast, patients who present with ASC and no indication for immediate surgery pose a greater dilemma. This group of patients includes those with medically refractory acute or chronic colitis. There are multiple biomarkers that have the potential to help inform the decision to pursue surgery in these settings (Table 1).

## 3. The Role of Biomarkers in Predicting Course of Disease and Need for Surgery

### 3.1. Determining the Course of the Disease; When Should We Operate?

Multiple biomarkers have been shown to predict disease course. Their capacity in this function has piqued interest as to whether they might also be associated with a need for surgical intervention and has occasionally resulted in their incorporation into predictive algorithms [20]. The ability to accurately determine which patients will ultimately require colectomy would be very helpful in counseling patients. This predictive capacity could also reduce the economic burden of the disease, as some studies have shown improved value with early surgery [21]. Although it is unlikely that any single test will become the standard for determining who will need surgery, there are several that may be helpful—especially in combination—in reducing prolonged unsuccessful medical treatment (Table 2).

### 3.2. Biomarkers

#### 3.2.1. C-Reactive Protein (CRP)

CRP is an acute-phase protein produced by hepatocytes and secreted as part of the systemic inflammatory response. It is rapidly and easily measured and has a relatively short half-life of 19 h. In healthy individuals, circulating levels are very low at <1 mg/L, but may increase several hundred-fold as part of an acute inflammatory reaction and can remain elevated over tenfold in chronic inflammatory conditions [22]. CRP has been recognized as a valuable component of the evaluation for UC for many decades [23]. An elevated CRP strongly suggests against a functional intestinal disorder, but has limited sensitivity and is not specific for UC [23]. Still, CRP has been recognized as a valuable component of the evaluation for UC for decades. Unfortunately, the CRP response in UC can be extremely heterogenous [22,23,24,25] (Table 3).

##### CRP in Predicting Short-Term Colectomy

The short half-life of CRP makes serial levels useful in assessing treatment response. In the acutely ill, hospitalized patient, persistently and significantly elevated CRP has been associated with steroid refractory disease and the need for surgery. In a study out of Sweden, among patients with UC, CRP elevation was significantly and independently associated with the need for colectomy within 30 days [26]. In that study, a CRP ≥25 mg/L and >4 stools daily on day 3 of IV corticosteroid therapy was predictive of the need for colectomy. The authors developed a fulminant colitis index calculated on day 3 of steroids (stools/day + 0.14 × CRP) which was later validated in a clinical trial [27]. The positive predictive value for colectomy within 90 days of a score ≥8 was 69%. Similarly, a study out of Oxford showed that 85% of patients with >8 bowel movements per day or 3–8 bowel movements per day and CRP >45 mg/L on day 3 required colectomy during that admission [28]. These studies predate the widespread use of variable biologic therapies. However, they do indicate that CRP may be a useful indicator of both disease severity and need for short term colectomy among patients with UC.

##### CRP in Predicting Long-Term Colectomy

CRP can also be useful at predicting the eventual need for colectomy in patients with UC. Henriksen, et al. showed that the odds of needing a colectomy is five times higher in UC patients with a CRP >23 mg/L at diagnosis, and three times higher in those with a CRP >10 mg/L one year after diagnosis [29]. Some suggest that CRP measurements above these thresholds should prompt a discussion with the patient about their risk for surgery and careful weighing of the possible consequences of delaying surgery against the potential benefits of additional medical treatment [20,29].

While CRP may be most helpful in the inpatient setting, it can also be utilized in long-term monitoring. For example, patients with ongoing elevation in CRP are likely to have persistent active disease and may need more aggressive treatment [30]. Unfortunately, CRP has not been shown to be predictive of relapse [31].

#### 3.2.2. Fecal Calprotectin and Lactoferrin

Fecal calprotectin (FC) is perhaps the most studied biomarker in UC/IBD [22,25,32,33,34,35,36,37,38,39,40,41,42,43,44,45,46,47]. It is a 36-kilodalton protein found in granulocytes and is stable in feces for several days [22,25,43,44]. It represents over 50% of cytosolic proteins in neutrophils, so its detection in stool is thought to correlate directly with neutrophil activation in a mucosal inflammatory response [48]. Testing is noninvasive, easy, and inexpensive and results return rapidly, making FC attractive as a potential biomarker. Several studies have shown that it is useful in determining which patients with symptoms of colitis warrant endoscopic analysis [22,25,34,42,43,46,47]. It is also more sensitive than CRP and allows for the evaluation of mild UC [43]. Limitations include a lack of specificity as it is elevated in other conditions that result in intestinal inflammation, including malignancy, infection, and polyps, as well as inconsistently defined cutoff values [24] (Table 3).

Fecal lactoferrin is very similar to calprotectin as it is a lysosomal protein found in neutrophils that is stable in stool for multiple days [25]. Levels have been shown to correlate with intestinal inflammation, though many studies indicate lower sensitivity and specificity than measuring FC [49]. As with FC, the fact that it is a component of neutrophils allows this marker to differentiate between inflammatory and functional intestinal disease.

Normalization of FC and lactoferrin levels has been shown to correlate with endoscopic response to treatment and mucosal healing [22,25,33,49]. It has also been shown to predict which patients will experience relapse and which might benefit from intensifying treatment [35,37,49]. Additional studies are required to determine how FC and lactoferrin levels in endoscopic response and relapse correlate with increased colectomy.

##### Fecal Calprotectin in Patients Requiring a Colectomy

Although much of the research regarding FC has been in the diagnosis and monitoring of UC, Ho et al. did study its ability to predict treatment failure in acute severe UC [50]. They found that FC levels were significantly higher in patients who required colectomy (1200 vs. 887 μg/g) and trended toward significance in distinguishing both steroid and infliximab nonresponders. Their analysis determined that a cutoff of 1922.5 μg/g was 97% specific for predicting the need for colectomy. Sensitivity was low at just 24%, though 87% of patients above that threshold ultimately had colectomy within the median follow-up of 1.1 years. Another study showed that FC levels >1500 μg/g on admission and >1000 μg/g on day 3 of IV corticosteroids was significantly associated with treatment failure and the need for rescue therapy or colectomy [51]. The association of admission levels >1500 μg/g and the requirement for colectomy in acute severe UC was confirmed by Wu et al. in 2019 [52]. These studies demonstrate the potential for FC to assist clinicians and patients considering colectomy. Still, additional study is warranted to determine the levels that optimally predict the need for colectomy.

#### 3.2.3. S100A12

S100A12 is a calcium-binding protein involved in various proinflammatory pathways and elevated levels have been demonstrated in inflammatory conditions [53,54]. When measured in stool, some studies have shown that S100A12 has higher sensitivity and specificity for distinguishing IBD from irritable bowel syndrome (IBS) than fecal calprotectin or lactoferrin [55] (Table 3).

Other studies have shown that serum levels are predictive of mucosal healing and correlate with disease activity [56,57]. It may therefore prove to be a more effective predictor of the need for colectomy than FC, but specific study of its utility for this purpose will be necessary.

#### 3.2.4. Fecal Myeloperoxidase

Another marker that reflects the activity of activated neutrophils in colonic mucosa is fecal myeloperoxidase (MPO). As with FC and fecal lactoferrin, it is stable in feces for several days [58]. MPO has been shown to be elevated in patients with UC compared to healthy controls, and MPO levels correlate with endoscopic grading of disease [59]. Likewise, MPO levels decrease in patients in remission, indicating an association with disease activity (Table 3).

MPO levels had a sensitivity of 89% but only 51% specificity in differentiating patients with UC from healthy controls. Another study showed superiority of fecal MPO measurement over FC in reflecting disease activity [60]. Like other markers, additional study will be required to determine its association with the need for colectomy.

#### 3.2.5. Serologies

##### Distinguishing Ulcerative Colitis from Crohn’s Disease

From a surgical standpoint, when considering a patient with IBD in nonemergent circumstances, the earliest and perhaps most critical objective is to determine whether a patient has UC or CD. From a medical standpoint, this distinction is less precarious, since many of the treatments are similar and switching from one treatment to another is standard. However, the surgical treatments of UC and CD are drastically different, and—in many cases—irreversible, highlighting the importance of diagnostic accuracy. Unfortunately, there is no gold-standard diagnostic test for UC [61]. It is largely a diagnosis of exclusion: If a patient clearly has IBD but lacks intestinal manifestations outside the colon/rectum and has no clear pathologic features of CD, we label it UC. Regrettably, many of the laboratory markers associated with UC—such as thrombocytosis, anemia, and elevated ESR—are nonspecific and minimally helpful in differentiating the diseases. Several serological biomarkers have shown promise and may support the process of differentiating patients with UC [25] (Table 2).

In general, Anti-*Saccharomyces cerevisiae* antibodies (ASCA) are associated with CD while perinuclear antineutrophil cytoplasmic antibodies (pANCA) are associated with UC, and the combination of these tests can distinguish between the two with 40–50% sensitivity and over 90% specificity [62]. Some newer serological tests have also been identified that might have a role in diagnosing IBD. In addition to ASCA, there are several other anti-glycan antibodies that have been investigated, including anti-laminaribioside carbohydrate IgG antibodies (ALCA), anti-chitobioside carbohydrate IgA antibodies (ACCA), anti-mannobioside carbohydrate IgG antibodies (AMCA), anti-laminarin (anti-L), and anti-chitin (anti-C). These have been found to have good specificity (75–99%) in distinguishing UC and CD from other gastrointestinal disorders but very limited sensitivity (11–40%) [63,64,65,66,67] (Table 3).

There are also antibodies against bacterial components that have been found to be common in patients with CD (about 50%), including those against *Eschericia coli* outer membrane porin C (anti-OmpC) and *Pseudomonas fluorescens-*associated sequence I2 (anti-I2) [68]. Anti-I2 was found to be positive in 54% of individuals with CD compared to 10% with UC, 19% with other inflammatory intestinal disease, and 4% of controls [69]. Some of these tests may be added to ASCA and pANCA to combine the sensitivity of distinguishing between UC and CD. Specifically, adding ALCA and ACCA to ASCA increases specificity to 85–99%, though there is a concomitant decrease in sensitivity from 66% to 27% [70]. Adding anti-OmpC, in contrast, does not help in distinguishing UC from CD [64].

Antibodies to flagellins (Anti-A4-Fla2 and Anti-Fla-X) have also been shown to have the capacity to distinguish between the two conditions. They were found to be positive in almost 60% of patients with CD and just 6% with UC [71] (Table 4).

Finally, a recent paper describes the use of a novel autoantibody against integrin αvβ6 in diagnosing UC [72]. Kuwada et al. found that this antibody was present in 92% of patients with UC and only 5% of patients with other pathologies (CD, diverticulitis, infections colitis, ischemic enteritis, Behçet’s disease, etc.). This translated to a sensitivity of 92.0% and specificity of 94.8%. Additionally, antibody titers correlated with disease activity, suggesting that this antibody may have potential for guiding treatment and escalation of therapy in addition to diagnosing UC. Additional study will be required to confirm its utility and to specifically determine its ability to distinguish between UC and other IBD.

Serological tests may be particularly valuable when considering the surgical care of the 10% of patients with diagnosis of inflammatory bowel disease unclassified (IBDU) or indeterminate colitis (IC), where a diagnosis is uncertain and patients may later develop pathognomonic signs or symptoms that allow for a confirmed diagnosis of either CD or UC. A prospective study of 97 patients diagnosed with IC found that 32% went on to receive a diagnosis of either CD or UC and that ASCA+/pANCA− was associated with the development of CD in 80% of patients [73]. In this study, ASCA−/pANCA+ serology was less specific, as 65% went on to develop UC while the remainder developed UC-like CD. In a similar study, Zhou, et al. demonstrated limited sensitivity for both tests (<50%), though they did show specificity >96% and high positive predictive values for their associated conditions [74]. These results suggest that in patients being evaluated for surgical treatment but with an uncertain diagnosis, testing serologies could assist in preventing inappropriate pouch creation by providing a more tailored approach.

##### Serologies to Predict Failure of Medical Therapy

Another potential application for serologies is in predicting failure of medical therapy. As previously discussed, ASCA and pANCA may help to guide appropriate surgical therapy in cases with an unclear diagnosis, but they have also been shown to have some ability to predict which patients will respond to infliximab therapy. One study has shown that pANCA seronegativity is significantly and positively associated with response to infliximab [75]. Another study showed that patients with UC and ASCA−/pANCA+ serology are less likely to have an early response to infliximab than those who are pANCA seronegative (OR 0.40) [76]. As this study included only five patients with ASC refractory to steroids, only one of whom required colectomy within 2 months, the true ability of serology to predict failure of rescue therapy will require further study.

#### 3.2.6. Drug-Related Biomarkers: Metabolite Levels, Drug Levels, and Antibodies

One method in which biomarkers might assist in determining the need for surgery is in predicting or monitoring the effect of treatment. Measuring drug or metabolite levels in the serum can ensure adequate dosing, which may prompt more rapid transition to alternative therapies or may suggest a need for surgery in nonresponders [25,77,78] (Table 3).

Additionally, use of biologic therapies can result in production of antidrug antibodies, which may be associated with relapse or the need to change treatments [79]. While these are helpful in guiding adjustments to therapy and counseling regarding surgery, a more efficient test would predict response prior to initiating therapy. Antimicrobial peptides (AMPs) and gut microbiota may perform in this manner for patients undergoing anti-TNF therapy. Proteomic analysis of pre-treatment biopsy specimens showed distinct patterns in patients for whom anti-TNF therapy was effective and those for whom it was not [80]. Additional study is needed to determine whether AMPs might also correlate with the need for colectomy and timing of surgical treatment.

#### 3.2.7. Peripheral Eosinophilia

Wright and Truelove described an association between eosinophils and UC in the 1960s, and the earliest case series date to the 1950s [81,82]. At that time, eosinophilia was thought to correlate with active disease [82]. More recent studies have discovered a role for peripheral blood eosinophilia (PBE) as a biomarker for disease activity [83,84]. This led Click, et al. to perform a registry analysis of 2066 IBD patients that correlated peripheral eosinophilia with a severe disease phenotype [85]. Among patients with UC, PBE was associated with extensive disease, active disease, PSC, aggressive medical treatment, higher healthcare utilization, hospitalization, and the need for surgery. Indeed, multivariate analysis showed that PBE was associated with hospitalization and surgery in UC, with adjusted ORs of 2.35 and 1.76, respectively. Further, time-to-event analysis showed that UC patients with PBE also had a significantly reduced time to colectomy. Additional studies confirming the predictive capacity of PBE with respect to surgical intervention will be integral to implementing this tool in decision making (Table 3).

#### 3.2.8. Serum Procalcitonin

Procalcitonin is a prohormone that is principally produced by non-neuroendocrine cells in response to systemic inflammation [86,87]. It has long been recognized as an important marker in sepsis; an association with UC was more recently identified [88]. Wu, et al. studied its correlation with outcomes in patients hospitalized with acute severe UC and found that admission procalcitonin levels were predictive of IV corticosteroid failure and the short-term need for surgery [52] (Table 3).

They also demonstrated a correlation with FC levels and that elevation of procalcitonin > 0.10 μg/L and FC > 1500 μg/g, predicted a need for colectomy in 86% of cases while 89% of patients with values <0.10 μg/L and <1500 μg/g responded to medical therapy [52]. This study suggests that serum procalcitonin, potentially in combination with FC levels, is promising as a biomarker predicting the need for colectomy.

#### 3.2.9. Hypoalbuminemia

While CRP levels increase as part of the inflammatory response, albumin functions as a negative acute phase reactant; levels drop during inflammation (due to both decreased synthesis and increased catabolism) [24,89]. Levels also decrease in response to conditions such as malnutrition and malabsorption, so that in UC it might reflect these downstream effects of inadequately controlled disease. Several studies have demonstrated that low levels of albumin, especially in patients presenting with ASC, are associated with refractoriness to corticosteroid therapy and with the need for surgery [90,91,92] (Table 3).

Ho, et al. proposed the use of a risk score incorporating hypoalbuminemia to identify patients who should have more aggressive approaches including earlier addition of second-line medical therapy or surgery [92]. More recently, a nationwide Veterans Affairs Hospitals study showed greater risk of needing any steroids, multiple courses of steroids, or second-line medical therapies in patients with hypoalbuminemia at diagnosis, as well as a trend toward increased need for colectomy [93]. While albumin may be effective as a component of a more comprehensive risk score, it is unlikely to be useful as an independent measure indicating need for colectomy.

#### 3.2.10. CRP/Albumin Ratio

Knowing that both CRP and albumin levels are associated with need for colectomy in patients with UC, Gibson, et al. proposed that the combination might have additional predictive capacity [94,95]. They found that an elevated CRP/albumin ratio on day 3 of IV corticosteroids was a more accurate marker of risk for colectomy within 30 days than was day 3 CRP or albumin alone. They also concluded that a CRP/albumin ratio greater than 0.85 optimally predicted need for colectomy within three years (50% of patients with day 3 CRP/albumin ratio above 0.85 required colectomy vs. 20% with a lower day 3 CRP/albumin ratio). Similarly, Choy, et al. sought to identify factors that might predict treatment failure and need for colectomy in a recent study [96]. While they were unable to confirm the early predictive value of CRP/albumin ratio, their analysis did show that a CRP/albumin ratio >0.37 at the time of discharge after treatment with infliximab (regardless of whether accelerated or standard induction was used) was significantly predictive of 12-month colectomy rates (Table 3).

#### 3.2.11. Neutrophil-to-Lymphocyte and Platelet-to-Lymphocyte Ratio

The neutrophil-to-lymphocyte ratio (NLR) and platelet-to-lymphocyte ratio (PLR) have been linked to outcomes in various disease processes [97,98,99,100]. Several studies in the last decade have shown some promise for NLR and PLR in differentiating patients with active or more severe UC from quiescent UC and healthy controls, though sensitivity and specificity have been limited [101,102,103,104]. They have also been shown to be able to predict disease response and relapse after medical therapies [105,106,107]. The optimal cutoff for these different applications has not been determined, but high NLR generally correlates with active disease, loss of response, and overall poor outcomes. These studies imply that NLR and PLR may be useful in predicting which patients will require surgical intervention, whether due to loss of response to medical therapy or more severe disease at presentation. However, optimal ranges and a direct correlation still need to be established before this can be used to predict colectomy in patients with UC (Table 3).

## 4. Conclusions

The prevalence of UC is increasing worldwide and treatment of UC is a significant burden on healthcare systems. The development of effective medical therapies has decreased the rates of surgery over time, but many patients still require surgical treatment, which also represents the only means of cure. Although patients who successfully undergo surgery experience good quality of life, delays in surgical treatment in the acute setting often result in increased morbidity and mortality. Delays for patients considering elective colectomy can result in increased costs of care and prolonged patient suffering. The ability to predict which patients ultimately require surgery could partially alleviate these encumbrances by informing discussions and decisions regarding timing of surgery.

Here we reviewed several biomarkers that show promise in this regard. Combining multiple biomarkers (such as CRP/albumin ratio and procalcitonin with fecal calprotectin) has resulted in improved accuracy in predicting which patients ultimately require surgery directly related to their disease. Further studies on the optimal use and combination of biomarkers and on the outcomes of earlier surgical intervention as dictated by such a predictive model are warranted to establish best practices.

## Figures and Tables

**Table 1 jcm-10-03362-t001:** Indications for surgery in ulcerative colitis and associated biomarkers.

Indication for Surgery	Biomarkers
Emergent condition (severe bleeding, toxic megacolon, etc.)	None currently (clinically determined)
Refractory disease (urgent or elective surgery)	C-Reactive Protein (CRP)
	Fecal calprotectin and lactoferrin
	S100A12
	Serologies
	Drug-related biomarkers
	Peripheral eosinophilia
	Procalcitonin
	Hypoalbuminemia
	CRP/Albumin ratio
	Neutrophil-to-Lymphocyte Ratio (NLR)
	Platelet-to-Lymphocyte Ratio (PLR)

**Table 2 jcm-10-03362-t002:** Biomarkers associated with UC and their utility in various areas.

Biomarker	Distinguish UC from CD	Predict IVCS Failure	Predict IFX Failure	Predict Long-Term Colectomy
CRP	-	+	+	+
Fecal calprotectin and lactoferrin	-	+	+	~
S100A12	-	~	~	~
Fecal myeloperoxidase	-	~	~	~
Serologies	+	~	+	~
Drug-related biomarkers	-	~	+	~
Peripheral eosinophilia	~	+	~	+
Procalcitonin	~	+	~	~
Hypoalbuminemia	-	+	+	+
CRP/Albumin ratio	-	+	~	+
NLR and PLR	-	~	+	~

+ indicates evidence demonstrating an association between the outcome and biomarker. - indicates evidence that the biomarker is not associated with the outcome. ~ indicates no evidence regarding association between the biomarker and the outcome. UC = Ulcerative colitis; CD = Crohn’s disease; IVCS = intravenous corticosteroids; IFX = infliximab; CRP = C-reactive protein; NLR = Neutrophil-to-lymphocyte ratio; PLR = Platelet-to-lymphocyte ratio.

**Table 3 jcm-10-03362-t003:** Applications and limitations of biomarkers studied in IBD.

Biomarker	Applications	Limitations
CRP	Distinguish UC from noninflammatory intestinal disorders Follow serial levels to assess treatment response May predict need for colectomy during admission for ASC May predict long-term need for colectomy	Not sensitive in UC Not specific for UC No discrimination between UC and CD
Fecal calprotectin and lactoferrin	Distinguish UC from noninflammatory intestinal disorders (more sensitive than CRP) Follow levels to assess treatment response/disease activity Correlates with mucosal healing May predict relapse May predict relapse May predict need for rescue therapy or colectomy	Not specific for UC Optimal cutoff points not defined Low sensitivity for predicting colectomy
S100A12	Distinguish UC from noninflammatory intestinal disorders (some studies show higher sensitivity than FC) Follow levels to assess treatment response/disease activity Correlates with mucosal healing	Not specific for UC Optimal cutoff points not defined Ability to predict treatment failure/colectomy not established
Fecal myeloperoxidase	Distinguish UC from noninflammatory intestinal disorders Follow levels to assess treatment response/disease activity	Not specific for UC Optimal cutoff points not defined Ability to predict treatment failure/colectomy not established
Serologies	May differentiate UC from CD Some ability to predict which patients with indeterminate colitis will develop UC vs. CD May predict early response to infliximab Some titers correlate with disease activity	Generally low sensitivity Ability to predict treatment failure/colectomy not established
Drug-related biomarkers	Ensure adequate dosing May prompt escalation in dosing or therapy or transition to alternative regimen May predict nonresponders to Anti-TNF drugs	Limited ability to predict response to treatment, particularly in ASC Ability to predict colectomy not established
Peripheral eosinophilia	Establishes more severe disease phenotype Associated with need for hospitalization and surgery Associated with reduced time to surgery	Ability to predict treatment failure not established Ability to predict short-term colectomy in ASC not established
Procalcitonin	Admission levels can predict IVCS failure in ASC May predict short-term need for colectomy in ASC, particularly when combined with FC	Ability to predict long-term colectomy in patients who initially respond not established
Hypoalbuminemia	May predict IVCS failure and colectomy in ASC Associated with need for multiple courses of steroids and second-line agents	Nonspecific Also a marker for malnutrition and increased surgical risk; patients may benefit from additional medical therapy
CRP/Albumin ratio	Day 3 value can predict IVCS failure and short-term colectomy in ASC Discharge levels after infliximab treatment can predict 12-month colectomy	Unclear when to measure Best clinical use not established
Neutrophil-to-lymphocyte ratio and platelet-to-lymphocyte ratio	Distinguish active from inactive UC Distinguish UC from healthy controls May predict mucosal healing with anti-TNF agents May predict loss of response to infliximab May predict relapse after tacrolimus-induced remission	Limited sensitivity and specificity Optimal cutoff points not defined Ability to predict need for colectomy not established

**Table 4 jcm-10-03362-t004:** Antibodies associated with Ulcerative Colitis and Crohn’s Disease.

Antibody	Prevalence in CD	Prevalence in UC
ASCA	+++	++
pANCA	+	+++
ALCA	++	+
ACCA	++	+
AMCA	++	+
Anti-L	++	+
Anti-C	++	+
Anti-OmpC	+++	+
Anti-I2	+++	+
Anti-A4-Fla2	+++	+
Anti-Fla-X	+++	+
Anti-integrin αvβ6	+	+++

+ indicates relatively low prevalence with studies showing peak prevalence under 25%; ++ indicates moderate prevalence with studies showing peak prevalence 25–50%; +++ indicates high prevalence with studies showing peak prevalence >50%. ASCA = anti-*Saccharomyces cerevisiae* antibodies; pANCA = perinuclear anti-neutrophil cytoplasmic antibodies; ALCA = antilaminaribioside carbohydrate antibodies; ACCA = antichitobioside carbohydrate antibodies; AMCA = antimannobioside carbohydrate antibodies; Anti-L = anti-laminarin antibodies; Anti-C = anti-chitin antibodies; Anti-OmpC = antibody to outermembrane porin C; Anti-I2 = antibody to *Pseudomonas fluorescens*-associated sequence I2; Anti-A4-Fla2 = antibody to flagellin A4-Fla2; Anti-Fla-X = antibody to flagellin Fla-X. Anti-integrin αvβ6 = antibody to integrin αvβ6.

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
