# Peer review of "The Role of Biomarkers in Surgery for Ulcerative Colitis: A Review"

_jcm, 2021, doi:10.3390/jcm10153362_

Round 1
Reviewer 1 Report
The authors reviewed the role of biomarkers in surgery for ulcerative colitis (UC). The article was well written, but the content was too greedy and difficult to read. The part of general statement was too long, not related to surgery. In addition, regarding distinguish UC from Crohn’ disease (CD), it is certainly related to the surgical procedure but not to the indication for surgery. This part (about distinguish) should be postponed until the last as supplementary information or deleted all.
The indications of surgery were broadly divided into emergency surgery, refractory disease (elective surgery), or the detection of high-grade dysplasia/cancer. The explanation of outline should be brief and described the biomarkers for each situation with Table.
Author Response
Please see the attachment
Biomarkers in Surgery for Ulcerative Colitis Review: Response to Reviewer 1 Comments
The authors would like to thank the reviewer for their thoughtful comments and suggestions. Please see below for a point-by-point response.
The authors reviewed the role of biomarkers in surgery for ulcerative colitis (UC). The article was well written, but the content was too greedy and difficult to read. The part of general statement was too long, not related to surgery. In addition, regarding distinguish UC from Crohn’ disease (CD), it is certainly related to the surgical procedure but not to the indication for surgery. This part (about distinguish) should be postponed until the last as supplementary information or deleted all.
The authors appreciate the reviewer’s recommendations. The indications for surgery in ulcerative colitis (UC) are multiple and complex. We have extensively edited the background information for brevity and clarity. Additionally, upon further reflection we determined that biomarkers related to the detection of colorectal cancer-while linked to UC-were outside the scope of this paper and we have removed a large portion of the material that was related to that topic. However, we do believe that distinguishing between UC and CD is essential in non-emergent surgery for UC. The operative management of the diseases is drastically different for urgent or elective indications. Therefore, one of the key questions a surgeon treating these conditions must ask is, “Am I sure of the diagnosis?” However, to better focus the paper, we shifted this discussion to the serologies section since this is the most relevant of the biomarkers in performing this function.
The indications of surgery were broadly divided into emergency surgery, refractory disease (elective surgery), or the detection of high-grade dysplasia/cancer. The explanation of outline should be brief and described the biomarkers for each situation with Table.
The reviewer’s point is well-taken. The background information has been edited for brevity. Additionally, as described above, the section related to dysplasia/cancer has been removed. A table has been added to summarize the indications for surgery and the associated biomarkers.

Reviewer 2 Report
In this review article, Matson et. al discuss the role of biomarkers in the medical and surgical management of Ulcerative Colitis.
I question the relevance of the section on "Background/Principles of Surgery for UC" in this paper (Section 2). Given the purpose of this manuscript (as evidenced by the title and conclusion) is to essentially review the role of biomarkers in predicting course of disease and need for surgery, this section detracts from that primary purpose. I would heavily edit this section for brevity, as Section 3 represents the main objective of the review.
Moreover, while it seems like the primary focus of this paper was supposed to be on UC, there is a significant focus on CD in the biomarkers section. In a similar vein to the previous comment, I think the authors need to better define the scope/purpose of this review article, and ensure that the data/results presented are in line with that. Overall, this seems like an IBD paper, rather than simply a UC paper. (All 4 tables talk about IBD, not just UC).
Finally, I also encourage the authors to outline how this study is novel, as there are numerous published manuscripts discussing the role of biomarkers in UC and CD.
Author Response
Please see the attachment
Biomarkers in Surgery for Ulcerative Colitis Review: Response to Reviewer 2 Comments
The authors would like to thank the reviewer for their thoughtful comments and recommendations. Please see below for a point-by-point response.
In this review article, Matson et. al discuss the role of biomarkers in the medical and surgical management of Ulcerative Colitis.
I question the relevance of the section on "Background/Principles of Surgery for UC" in this paper (Section 2). Given the purpose of this manuscript (as evidenced by the title and conclusion) is to essentially review the role of biomarkers in predicting course of disease and need for surgery, this section detracts from that primary purpose. I would heavily edit this section for brevity, as Section 3 represents the main objective of the review.
The authors appreciate this suggestion and have extensively edited Section 2 for brevity and clarity. Additionally, Section 4 on dysplasia and colorectal cancer has been removed to better focus on our primary objective of reviewing the role of biomarkers in guiding the decision to pursue and timing of surgery in ulcerative colitis.
Moreover, while it seems like the primary focus of this paper was supposed to be on UC, there is a significant focus on CD in the biomarkers section. In a similar vein to the previous comment, I think the authors need to better define the scope/purpose of this review article, and ensure that the data/results presented are in line with that. Overall, this seems like an IBD paper, rather than simply a UC paper. (All 4 tables talk about IBD, not just UC).
The authors would like to thank the reviewer for this comment. The focus of this article is on the role that biomarkers play in determining the need and timing for surgery in ulcerative colitis (UC). Because surgical management of UC is drastically different than operative management of Crohn’s disease (CD) in anything other than the emergent setting, we felt that the role of biomarkers in distinguishing between the two entities was an important part of this paper. However, we have reorganized and significantly edited the material to primarily focus on UC, including removal of most of the generic background information on surgery as well as the section on biomarkers in cancer. We have also moved most of the discussion of distinguishing between UC and CD to the section on serologies, as those are the most relevant biomarker in this regard.
Finally, I also encourage the authors to outline how this study is novel, as there are numerous published manuscripts discussing the role of biomarkers in UC and CD.
The authors appreciate the importance of this question. In our search of the existing literature, we found numerous manuscripts about various biomarkers in UC and CD. Indeed, there are multiple published review articles describing many of these biomarkers. However, we did not identify any previously published reviews specifically focusing on the role of biomarkers in the surgical management of UC. We believe that this article can serve as a consolidated resource of the existing biomarkers that have relevance to surgical management of UC and may help to drive the specific exploration of these and novel biomarkers in determining the need and timing for surgery in UC. We rephrased our thesis statement at the end of the introduction to reflect this.

Round 2
Reviewer 1 Report
Thank you for your response. I think the article has become much succinctly.
If the authors are particular about distinguishing between UC and CD, why not cite the following recent article. Kuwada T, et al. Gastroenterology. 2021;160:2383-2394.
Reviewer 2 Report
I appreciate the edits the authors have made to the manuscript. In this revised manuscript, the thesis of the review describes "examining the role of biomarkers in determining the need for and timing of surgery in UC" (Introduction). However, throughout the manuscript, most of the discussion centers on how these biomarkers can be used to predict the need for surgery, rather than the exact timing with with surgery should occur. I would encourage the authors to modify either this statement, or the Discussion, in accordance with their overall purpose.
